# Prediction of sub-pyramid texturing as the next step towards high efficiency silicon heterojunction solar cells

Feihong Chu[1,4], Xianlin Qu[2,4], Yongcai He[1,3], Wenling Li[1], Xiaoqing Chen [1], Zilong Zheng [1] ✉, Miao Yang[3], Xiaoning Ru[3], Fuguo Peng[3], Minghao Qu[3], Kun Zheng [1] ✉, Xixiang Xu [3] ✉, Hui Yan[1] & Yongzhe Zhang [1] ✉

The interfacial morphology of crystalline silicon/hydrogenated amorphous silicon (c-Si/a-Si:H) is a key success factor to approach the theoretical efficiency of Si-based solar cells, especially Si heterojunction technology. The unexpected crystalline silicon epitaxial growth and interfacial nanotwins formation remain a challenging issue for silicon heterojunction technology. Here, we design a hybrid interface by tuning pyramid apex-angle to improve c-Si/a-Si:H interfacial morphology in silicon solar cells. The pyramid apex-angle (slightly smaller than 70.53°) consists of hybrid $(111)_{0.9}/(011)_{0.1}$ c-Si planes, rather than pure (111) planes in conventional texture pyramid. Employing microsecond-long low-temperature (500 K) molecular dynamic simulations, the hybrid (111)/(011) plane prevents from both c-Si epitaxial growth and nanotwin formation. More importantly, given there is not any additional industrial preparation process, the hybrid c-Si plane could improve c-Si/a-Si:H interfacial morphology for a-Si passivated contacts technique, and wide-applied for all silicon-based solar cells as well.

Over the past few decades, silicon wafer-based silicon solar cells have dominated the photovoltaic (PV) industry, given low production cost, high energy-conversion efficiency and long-term stability[1-4]. The silicon heterojunction (SHJ) as state-of-the-art technology, is made of silicon wafers employing hydrogenated amorphous silicon (a-Si:H) passivated contacts for both polarities based on i/n and i/p stacks of thin-film. In consideration of high open circuit voltage ($V_{OC}$), SHJ is one of the most promising technologies for achieving high efficiency[5,6], and the theoretical power conversion efficiency (PCE) of SHJ solar cells is 28.6%[7]. Recently, LONGi has received 26.8% of world record efficiency, based on SHJ manufacturing improvements over the years[8]. The interfacial morphology of the crystalline silicon (c-Si)/a-Si:H heterojunction is a key success factor in further SHJ efficiency improvement[9]. The a-Si with continuous random network structure, prefers to convert to a well-ordered crystal lattice configuration, and

the latter has face-centered cubic and hexagonal close-packed structures; therefore, the c-Si epitaxy and embedded nanotwins are generated at the c-Si/a-Si:H interface[10,11]. However, both of them have a rough interface, which could introduce the deep-level defect states with additional charge carrier recombination and nonradiative open circuit voltage losses ($\Delta V_{OC}$)[11]; meanwhile, the twin boundary scattering enlarge the effective mass along charge transport direction, and result in low charge carrier mobility[12,13]. In order to prevent from those interfacial damages, the low-temperature (around 500 K) a-Si deposition technology was developed as the passivation process for the high-performance SHJ solar cells[14,15]. Although, the epitaxial c-Si and nanotwins are hardly prevented at the c-Si/a-Si:H interface[10,11], we noticed that an intrinsic a-Si deposition ultra-thin (<1 nm) buffer layer formed by radio-frequency plasma enhanced chemical vapor deposition (RF-PECVD) could lower the formation of c-Si epitaxy and

[1]Faculty of Materials and Manufacturing, Faculty of Information Technology, Beijing University of Technology, Beijing, China. [2]Center for Microscopy and Analysis, Nanjing University of Aeronautics and Astronautics, Nanjing, China. [3]LONGi Central R&D Institute, Xi'an, China. [4]These authors contributed equally: Feihong Chu, Xianlin Qu. ✉e-mail: zilong.zheng@bjut.edu.cn; kunzheng@bjut.edu.cn; xuxixiang@longi.com; yzzhang@bjut.edu.cn

nanotwins in previous report[11]. Unfortunately, it may bring time-consuming and additional industrial preparation process. On the other hand, our previous studies indicated the epitaxial growth can be suppressed by a hybrid interface with atomic steps as well, which deviates from the conventional c-Si (111) plane[11]. Therefore, it is urgent to understand the nature of the hybrid interface, which could provide a potential approach to further improve c-Si/a-Si:H interfacial morphology and the performance of SHJ solar cells.

At the c-Si-a-Si:H interface, both density and arrangement of Si atoms are dependent on the c-Si lattice orientation, which lead to anisotropic epitaxial growth property. Low-index crystal planes with high-symmetry structure are energy stabilized interface, for instance, (100), (011), and (111) planes. New epitaxial Si atoms at (111) plane are not energy stable, because of three dangling bonds exposure, rather than two dangling bonds at (100) and (011) planes. Similarly, (100) and (011) planes with two dangling bonds show higher etch rate than that of (111) plane with one dangling bond. Therefore, the conventional c-Si/a-Si:H interface is always based on pure c-Si (111) plane, given the relatively low epitaxial growth rate and high etch rate. In this work, in combination of ex- and in-situ high resolution transmission electron microscope (HRTEM) images of high-efficiency (PCE = 24.85%) SHJ industrial devices, a new modeling of c-Si/a-Si:H interfacial morphology was obtained. We then performed microsecond-long (-μs) low-temperature (500 K) molecular dynamic (MD) simulations, and proposed a strategy for a-Si passivated contacts technique, that a hybrid interface with the mixture of those low-index planes could reduce both interfacial c-Si epitaxy and nanotwins.

The pyramid texture of SHJ solar cells presents great anti-reflective performance. The standard apex-angle of pyramid is 70.53°, which is between the (111) surface and the (1$\bar{1}$1) surface, conventionally. Yan et al. investigated the impact of the apex-angle on the reflectance; as the apex-angle decreasing from 130° down to 50°, the reflectance could reduce from 36.7 to 5.5%[16]. Actually, the small apex-angle works in favor of multiple light absorption, and result in low reflection. By tuning apex-angle of pyramid, the hybrid interfacial surface can consist of (111)/(100) c-Si plane, as well as, (111)/(011) c-Si plane, respectively. While the former's apex-angle is larger than 70.53°, and the latter one is smaller than 70.53°. In this work, we investigated interfacial c-Si epitaxy and nanotwin growth at traditional pure (111)

plane, hybrid (111)/(100) plane, and hybrid (111)/(011) plane based on all-atom MD simulations. The calculated nanotwins distribution has a good agreement with measured atomic-resolution in-situ HRTEM characterization. The hybrid interface with a fraction (x), for instance, $(111)_{1-x}/(011)_x$, significantly improves the c-Si/a-Si interfacial morphology with low c-Si epitaxy and nanotwins. Therefore, we design a hybrid interface by tuning pyramid apex-angle, which could prevent from both epitaxial growth and nanotwin formation in SHJ solar cells. More importantly, without additional industrial preparation process, the hybrid interface could improve c-Si/a-Si:H interfacial morphology for a-Si passivated contacts technique, and wide-applied for the silicon-based (single-junction and tandem) solar cells in the future.

## Results

### Interfacial morphology on hybrid (111)/(100) c-Si plane

In order to further understand the phenomenon that the atomic steps on c-Si/a-Si:H interface can reduce nanotwins following our previous work[11], the c-Si/a-Si:H interfaces were fabricated using the same industry-compatible process as that with high-efficiency (PCE = 24.85%) SHJ solar cells, and many large apex-angles (>70.53°) of pyramid were obtained, see Fig. 1b, c. The schematic diagram of the cross-sectional structures of the SHJ solar cells is in Fig. 1a. In order to investigate the interface configuration, we fabricated TEM samples by employing the focused ion beam (FIB) technique. The cross-section of pyramid structure was marked with the red rectangular area in Fig. 1b. Figure 1c is the low magnified TEM image of the pyramid, where the apex angle θ is 74.0°. The conventional θ is known as 70.53°, which is between the standard (111) and standard (1$\bar{1}$1) planes of c-Si. When the apex angle θ is larger than 70.53°, the increased angle results in c-Si surface deviating from (111) plane and presenting a series of atomic steps. Actually, the nature of the atomic steps is hybrid c-Si plane configuration at the c-Si/a-Si:H interface, rather than pure (111) c-Si plane; the former one consists of two lattice c-Si planes, the major one is still the (111) facet, and the minor one is (100) facet. Then, we further investigated the structural characteristics of the c-Si/a-Si:H interface. Figure 1d showed morphology on flat (111) plane interface, where high density nanotwins and c-Si epitaxial with a thickness of 1-2 nm were observed. Figure 1e presented the morphology on hybrid (111)/(100) plane interface, while, in comparison with pure flat (111) plane, there

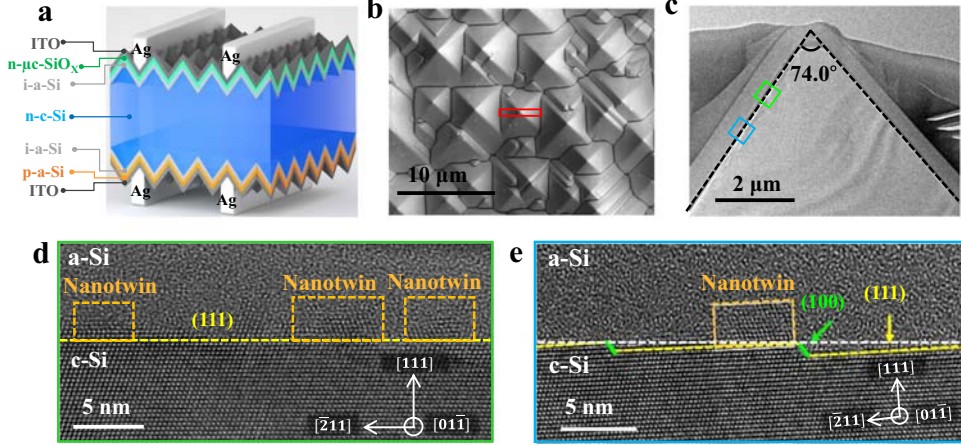

**Fig. 1 | c-Si/a-Si:H interface structure characteristics of SHJ solar cell. a** Cross-section schematic image of the SHJ solar cell, from top to bottom are Ag (silver electrode), ITO (indium tin oxide), n-μc-SiO$_X$ (n-type microcrystal SiO$_X$), i-a-Si (intrinsic a-Si), n-c-Si (n-type crystal Si), i-a-Si (intrinsic a-Si), p-a-Si (p-type a-Si), ITO and Ag. **b** SEM image showed the morphology of pyramid structure on the surface of SHJ solar cells. **c** Low-magnification TEM image showed the cross-section of pyramid structure marked by red rectangle in Fig. 1b. The green rectangle marked

the flat (111) c-Si plane and the blue rectangle marked hybrid c-Si plane (111)/(100) with the atomic steps, respectively. **d, e** HRTEM images viewed from the [01$\bar{1}$] orientation presented the c-Si/a-Si:H interface with the flat (111) plane (**d**) and the atomic steps (**e**), while the related area marked by green rectangle and blue rectangle in Fig. 2c, respectively. The yellow dash line represented the (111) plane, the green dash line represented the (100) plane; the white dashed line represented c-Si surface and the orange rectangle represented the nanotwins.

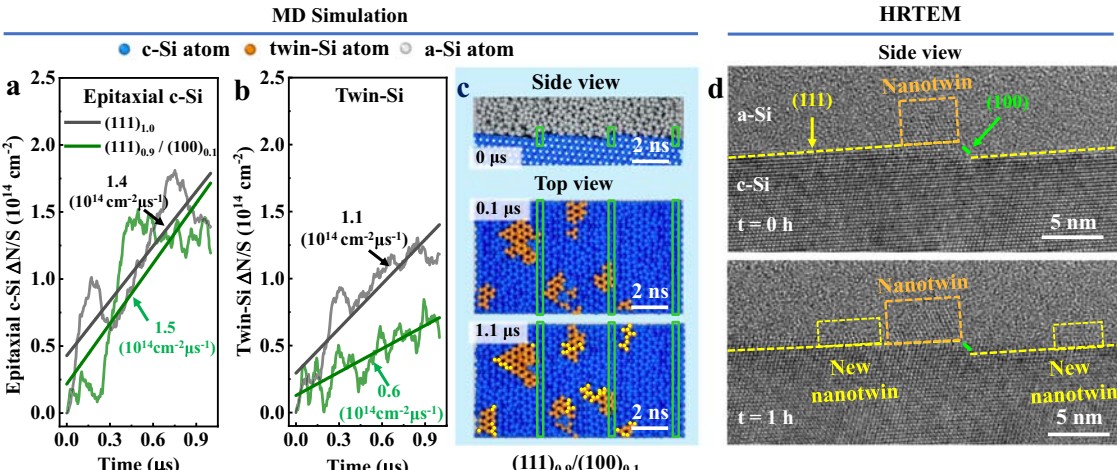

**Fig. 2 | Molecular dynamic simulations and in-situ HRTEM images of c-Si/a-Si interface on pure (111) c-Si plane and hybrid $(111)_{0.9}/(100)_{0.1}$ c-Si plane. a, b** The generations of epitaxial c-Si (**a**) and twin-Si atoms (**b**) as a function of time. **c** Interfacial morphological on hybrid $(111)_{0.9}/(100)_{0.1}$ plane. The side view (top panel) c-Si/a-Si interface of initial hybrid $(111)_{0.9}/(100)_{0.1}$ plane; top view of c-Si/a-Si interface of hybrid $(111)_{0.9}/(100)_{0.1}$ plane, following 0.1 μs (middle panel) and 1.1 μs (bottom panel) molecular dynamic simulations with low-temperature (500 K). The blue balls represented c-Si, the orange balls represented the initial twin-Si, and the white balls represented a-Si. In order to observe the distributions of epitaxial c-Si and nanotwin, a-Si atoms are hidden on top view, new generated twin-Si atoms are marked with yellow circles, the green rectangle represented the (100) component. **d** In situ HRTEM images along the $[01\bar{1}]$ direction showed the c-Si/a-Si:H interface before (top panel) and after (bottom panel) heating at 500 K for 1 h. The orange dash rectangles are nanotwins, the yellow dash rectangles are new generated nanotwins.

are low-density nanotwins nearby atomic steps. The nanotwins prefer to grow on the (111) component rather than the (100) component[17]. Furthermore, these similar low-density nanotwins distribution at interface were observed, see Fig. S1 in Supporting Information (SI).

## The c-Si epitaxy and nanotwins growth on hybrid (111)/(100) c-Si plane

During the preparation of SHJ solar cells, the process could promote the formation of both c-Si epitaxy and nanotwin at the c-Si/a-Si:H interface. Therefore we investigated their distributions for both a-Si depositing process and post-deposition time evolution status, by employing molecular dynamics (MD) simulations with low-temperature (500 K) as industrial preparing process[11,15]. The modeling initial configurations of c-Si/a-Si interface are based on HRTEM images of high-efficiency (PCE = 24.85%) SHJ industrial devices, as shown in Fig. S2 in SI[11].

The interface with single (111) c-Si plane started forming nanotwin structure at 0.15 μs, and the latter meets to form embedded nanotwin at 0.9 μs, as shown in Fig. S3 in SI. Considering the time scale for the formation of embedded nanotwins, we simulated the morphological evolution of the c-Si/a-Si interface at 500 K with a total time of 1 μs. According to anisotropy of c-Si epitaxy and nanotwin on different low-index surfaces (see details in Supplementary Note 1), the hybrid plane can be regarded as consisting of several low-index surface compositions, and the latter have a critical role in the impact on the epitaxial c-Si and nanotwin[17,18]. The new generated epitaxial c-Si atoms as a function of time, as shown in Fig. 2a. A linear increasing trend of c-Si epitaxial growth was obtained, and the epitaxial c-Si generation rate were estimated with a line-fitting, see the slope of fitting curves in Fig. 2a. The hybrid $(111)_{0.9}/(100)_{0.1}$ plane has a similar c-Si epitaxy rate $(1.5 \times 10^{14} \text{ cm}^{-2} \cdot \mu s^{-1})$ as that of the pure (111) plane $(1.4 \times 10^{14} \text{ cm}^{-2} \cdot \mu s^{-1})$. It indicated a small amount of (100) component cannot have a significant effect on c-Si epitaxy. However, when traditional (111) plane mixing with (100) plane, the nanotwin growth rate was dropped down significantly, from $1.1 \times 10^{14} \text{ cm}^{-2} \cdot \mu s^{-1}$ at pure (111) plane down to $0.6 \times 10^{14} \text{ cm}^{-2} \cdot \mu s^{-1}$ at the hybrid $(111)_{0.9}/(100)_{0.1}$ plane, see Fig. 2b. Compared with interface defect density at c-Si/a-Si:H of high-efficiency SHJ devices (about $10^{10} \text{ cm}^{-2}$)[19,20], the interfacial epitaxial c-Si and

nanotwin atomic density is several orders of magnitude more than interface defect density, in consideration of the nanotwin or epitaxial c-Si consisting of thousands of Si atoms only introduce a few defect states.

Moreover, we further investigated the spatial distribution of nanotwins at c-Si/a-Si interface of hybrid $(111)_{0.9}/(100)_{0.1}$ plane, see Fig. 2c; while the top panel (side view) presented the initial modeling structure of the c-Si/a-Si interface along the <110> direction; the flat area is (111) facet, and the atomic step area is (100) facet; the latter were marked with green rectangles. In the equilibrium configurations exported from MD simulations with a short time-scale of 0.1 μs, the nanotwins (marked with orange color) are generated and mainly localized on the (111) facet rather than (100) facet, see Fig. 2c middle panel. The nanotwins grow until the edge of (111) facet, and blocked by the (100) facet of atomic steps at c-Si/a-Si interface. The blocking effect on hybrid (111)/(100) plane was confirmed by HRTEM imaging, see Fig. 2d top panel; where the nanotwins grew only on (111) facet, as shown in orange dashed rectangle, but cannot pass across the boundary between (100) and (111) facet. In order to further confirm the (100) nanotwin blocking effect, we made a MD simulation with a long time-scale of 1.1 μs. The new generated nanotwins (marked with yellow color) still distribute only on (111) facets, and the (100) facets prevent from the nanotwins growth, see Fig. 2c bottom panel. Employing in situ heating microelectromechanical systems (MEMs) at 500 K for 1 h, the atomic-resolution HRTEM images of hybrid (111)/(100) plane demonstrated the same conclusion as theoretical results, that new growing nanotwins (marked by the yellow dashed rectangle) were blocked by the (100) facet of atomic steps at c-Si/a-Si:H interface, see Fig. 2d bottom panel.

## The c-Si epitaxy and nanotwin growth on hybrid (111)/(011) c-Si plane

The above hybrid (111)/(100) c-Si plane significantly inhibits the nanotwins growth, unfortunately, the c-Si epitaxy cannot be suppressed. Compared with the property of pure (100) plane, the pure (011) plane could prevent from both nanotwin growth and c-Si epitaxy. By control the ratio of between KOH and isopropyl alcohol in etching solution of texturing process, the etching rate of (100) and (011) c-Si

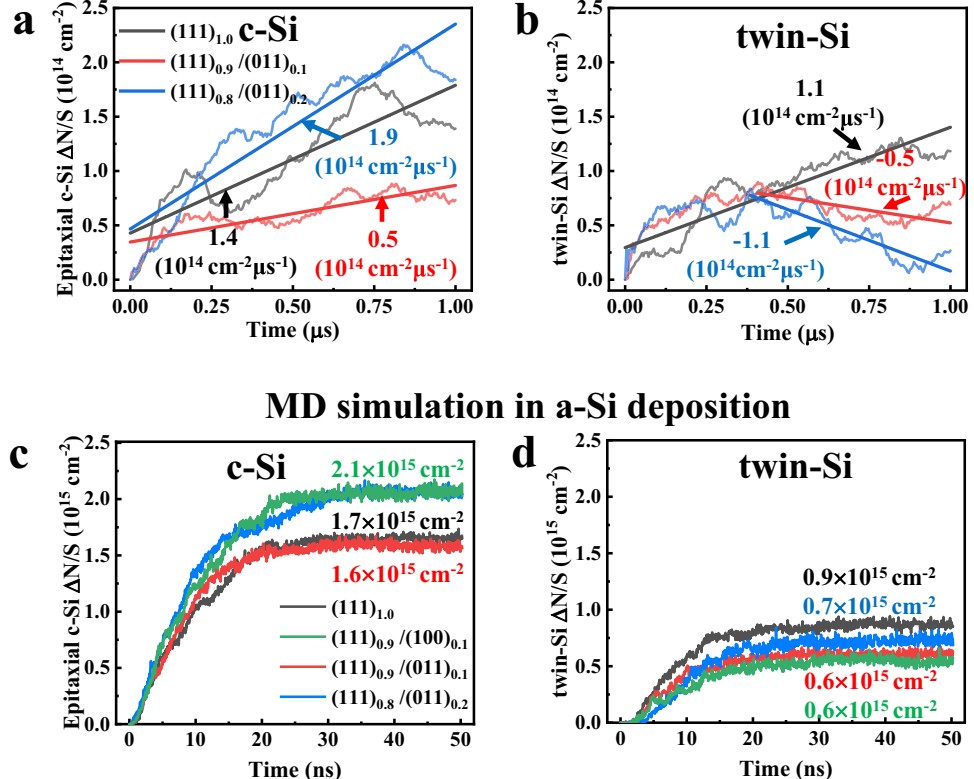

**Fig. 3 | Molecular dynamic simulations of epitaxial c-Si and twin-Si growth.**
**a**, **b** The densities of new generated epitaxial c-Si (**a**) and twin-Si (**b**) on pure (111) c-Si plane, hybrid $(111)_{0.9}/(011)_{0.1}$ plane, and hybrid $(111)_{0.8}/(011)_{0.2}$ plane, as a function of time at 500 K. The slopes of fitting curves represent the generating rates. **c**, **d** In the simulations of amorphous Si deposition, the densities of epitaxial c-Si (**c**) and twin-Si growth (**d**) on pure (111) c-Si plane, hybrid $(111)_{0.9}/(100)_{0.1}$ plane, hybrid $(111)_{0.9}/(011)_{0.1}$ plane, and hybrid $(111)_{0.8}/(011)_{0.2}$ plane, as a function of time.

plane could be adjusted; while it is possibly to obtain a new hybrid c-Si plane, (111)/(011) interface, rather than above (111)/(100) plane, and the former has a pyramid with apex angle $\theta < 70.53°$[21]. We therefore investigated the property of (111)/(011) c-Si plane, in order to look for better suppression capability to nanotwin growth and c-Si epitaxy, compared to traditional (111) plane or hybrid (111)/(100) c-Si plane.

The modeling hybrid $(111)_{0.9}/(011)_{0.1}$ plane was shown in Fig. S2, and we performed microsecond-long MD simulation for c-Si/a-Si interfacial morphology with low-temperature (500 K), which is the same as industrial a-Si deposition technology. The c-Si epitaxial growth rate ($0.5 \times 10^{14}$ cm$^{-2}$ · μs$^{-1}$) of hybrid $(111)_{0.9}/(011)_{0.1}$ plane is significantly lower than both of traditional (111) plane ($1.4 \times 10^{14}$ cm$^{-2}$ · μs$^{-1}$) or hybrid $(111)_{0.9}/(100)_{0.1}$ plane ($1.5 \times 10^{14}$ cm$^{-2}$ · μs$^{-1}$); while, the former with (011) facet atomic steps can suppress c-Si epitaxy effectively, as shown in Fig. 3a. However, when we continue to increase the (011) facet component (x) to 0.2, the hybrid $(111)_{0.8}/(011)_{0.2}$ c-Si plane increase the c-Si epitaxial rate slightly ($1.9 \times 10^{14}$ cm$^{-2}$μs$^{-1}$) compared with $1.4 \times 10^{14}$ cm$^{-2}$ · μs$^{-1}$ of pure (111) plane. More component of (011) facet cannot be helpful, considering high the c-Si epitaxial rate of pure (011) plane[22]. The c-Si/a-Si interfacial energies were exported from MD simulations, and the lowest one is on hybrid $(111)_{0.9}/(011)_{0.1}$ c-Si plane, compared with the three others on traditional (111), hybrid $(111)_{0.8}/(011)_{0.2}$ and above hybrid $(111)_{0.9}/(100)_{0.1}$ c-Si planes, see Fig. S4 in SI; while it implied the former is energetic stable. Moreover, when a-Si atoms converting to c-Si atoms, the averaged stabilization energies on each atom in the system, were obtained with the order of 8 meV on hybrid $(111)_{0.9}/(011)_{0.1}$, 11 meV on hybrid $(111)_{0.9}/(100)_{0.1}$, 12 meV on traditional (111), and 14 meV on hybrid $(111)_{0.8}/(011)_{0.2}$; while hybrid $(111)_{0.9}/(011)_{0.1}$ plane provided the minimal driving force for both epitaxial c-Si and twin silicon (twin-Si) growth.

We next looked for the twin-Si growth on a-Si/$(111)_{0.9}/(011)_{0.1}$ c-Si interface. Since there are not any twin-Si atoms on initial modeling configurations, they started growing on both traditional (111) and hybrid (111)/(011) planes, following the first 0.4 μs MD simulations. While, those twin-Si atoms are mainly localized on (111) area, which is the major component on both pure (111) and hybrid (111)/(011) planes, and twin-Si growth has the same rate around $1.1 \times 10^{14}$ cm$^{-2}$ · μs$^{-1}$ at the beginning. However, when the twin-Si atoms extend to the (011) component on hybrid (111)/(011) plane, they start reducing with the rate of $-0.5 \times 10^{14}$ cm$^{-2}$ · μs$^{-1}$ and $-1.1 \times 10^{14}$ cm$^{-2}$ · μs$^{-1}$ on hybrid $(111)_{0.9}/(011)_{0.1}$ plane and $(111)_{0.8}/(011)_{0.2}$ plane, respectively. The negative twin-Si growth rates are due to the low potential barrier (0.10 eV) of transformation from twin-Si to c-Si on (011) area, compared with 0.34 eV barrier on (111) area, which are calculated via density functional theory (DFT) method, see Fig. S5 in SI. The twin-Si atoms disappear mainly around the (011) plane, and considering more component on $(111)_{0.8}/(011)_{0.2}$ plane, the disappearance of twin-Si was obvious rather than on hybrid $(111)_{0.9}/(011)_{0.1}$ plane, and see Fig. S6 in SI.

As the above (100) twin-Si blocking effect on hybrid (111)/(100) plane, the (011) component presents the same blocking function on (111)/(011) plane. When the latter stay the same, for instance (011) component keep 0.1 on $(111)_{0.9}/(011)_{0.1}$ plane, the breadth of each (011) atomic steps impact the twin-Si blocking effect. When the (011) breadth become large, each the related (111) continuous region are expanded as well, see Fig. S7 in SI. Based on 1.5 μs MD simulations with modeling structures of multiple two-atom breadth of (011) steps, the twin-Si distributions were significantly increased, compared with the result of two-atom breadth on the hybrid $(111)_{0.9}/(011)_{0.1}$ plane, as shown in Fig. S8 in SI. Therefore, the short breadth of (011) atomic steps may prevent from twin-Si growth region expansion effectively,

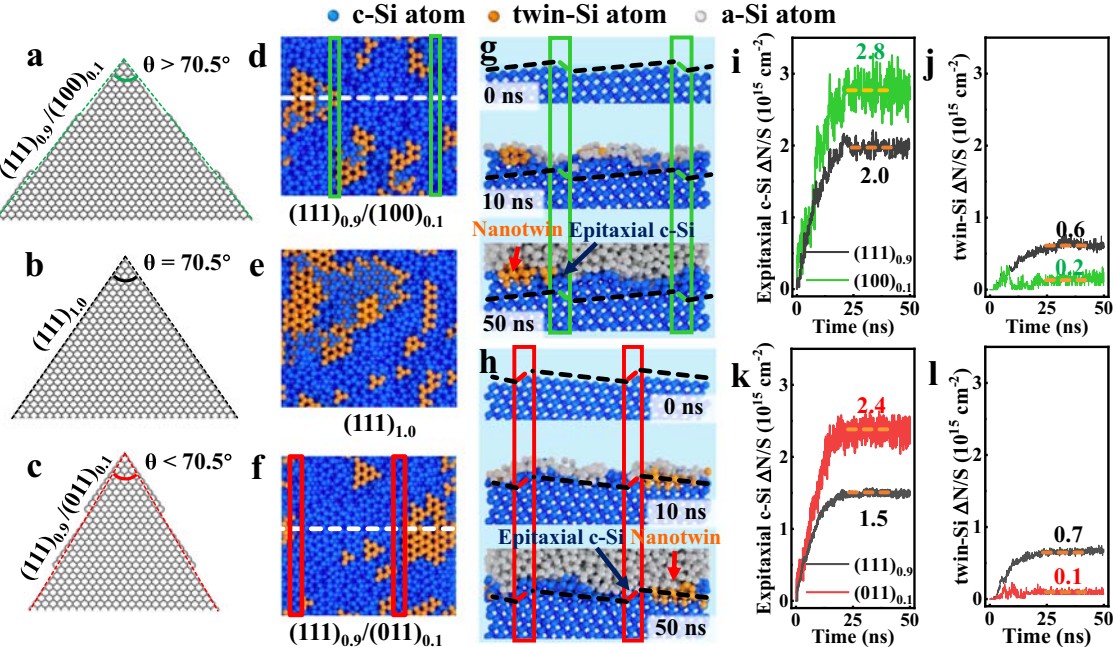

**Fig. 4 | Interfacial morphology of molecular dynamic simulations in a-Si deposition process. a–c** Schematic diagram of textured pyramid with apex angle $\theta > 70.53°$ (**a**), $\theta = 70.53°$ (**b**) and $\theta < 70.53°$(**c**). **d–f** the top view of the c-Si/a-Si interface for the hybrid $(111)_{0.9}/(100)_{0.1}$ plane (**d**), pure (111) plane (**e**), and hybrid $(111)_{0.9}/(011)_{0.1}$ plane (**f**), following a-Si deposition simulations, while in order to present the distribution of epitaxial c-Si and nanotwins, all a-Si atoms are hidden; the green and red rectangle represents the minor plane in hybrid c-Si interface, (100) facet and (011) facet, respectively; the white dashed line represents the side view position. **g, h** The side view of c-Si/a-Si interface of hybrid $(111)_{0.9}/(100)_{0.1}$ plane (**g**) and hybrid $(111)_{0.9}(011)_{0.1}$ plane (**h**) along with time $t = 0$ ns (top panel), $t = 10$ ns (middle panel), as well as $t = 50$ ns (bottom panel). The black dash line represents (111) c-Si facet, the green rectangle and dash line represent the (100) c-Si facet, the red rectangle and dash line represent the (011) c-Si facet. **i–l** The epitaxial c-Si (**i, k**) and twin-Si (**j, l**) of (111) component (black curves), (100) component (green curves), and (011) component (red curves) as a function of time, on hybrid $(111)_{0.9}/(100)_{0.1}$ plane (**i, j**) and hybrid $(111)_{0.9}/(011)_{0.1}$ plane (**k, l**).

via enhancing the (011) twin-Si blocking effect on the hybrid $(111)_{0.9}/(011)_{0.1}$ plane.

## The formations of both epitaxial c-Si and twin-Si in a-Si deposition

In addition to the dynamic evolution process at c-Si/a-Si interface in SHJ solar cells, we investigated c-Si epitaxial growth and twin-Si formation in the a-Si deposition passivated process as well. The c-Si/a-Si interfacial morphologies of traditional (111) plane, hybrid (111)/(100) plane, and hybrid (111)/(011) plane were obtained via MD simulations under the a-Si deposition condition, see Fig. 4e, d and f, respectively. While the new generated nanotwins (marked with orange color) on the former are much more than those on hybrid planes. Figure 3c, d presented the atomic number of new epitaxial c-Si and nanotwins as a function of time. All of them have levelled off following a period of rapid linear rise. The reason is that, for instance, the thickness of deposited a-Si atoms is around 2 nm after 30 ns MD simulations, which prevented the new depositing a-Si atoms from touching the c-Si layer (or arriving at the c-Si/a-Si interface). Therefore, the amounts of both generated epitaxial c-Si and nanotwins reached stable statuses, and we exported those numbers from the MD simulations, which were marked in Fig. 3c, d as well. The fewest two epitaxial c-Si reached $1.6 \times 10^{15}$ cm$^{-2}$ on hybrid $(111)_{0.9}/(011)_{0.1}$ plane and $1.7 \times 10^{15}$ cm$^{-2}$ on traditional (111) plane. While, the fewest two twin-Si formation reached both $0.6 \times 10^{15}$ cm$^{-2}$ on hybrid $(111)_{0.9}/(100)_{0.1}$ plane or hybrid $(111)_{0.9}/(011)_{0.1}$ plane. Considering the lower formation energy of c-Si rather than that of twin-Si, see above Fig. S5 in SI, the epitaxial c-Si growth is more than twin-Si formation. In consistent with the above result of c-Si/a-Si interface in SHJ solar cells, the hybrid $(111)_{0.9}/(011)_{0.1}$ plane should be considered as a key candidate to reduce both epitaxial c-Si and nanotwin-Si based on the a-Si deposition MD simulations.

We then obtained the dynamic snapshots for both hybrid $(111)_{0.9}/(100)_{0.1}$ plane or hybrid $(111)_{0.9}/(011)_{0.1}$ plane, see Fig. 4g and h, respectively. Both (100) component and (011) component prevented from twin-Si expansion along the c-Si/a-Si interface, as well as, have the role of twin-Si blocking function on hybrid (111)/(100) and hybrid (111)/(011) c-Si planes. We derived both epitaxial c-Si and twin-Si atom areal density on each (111), (100), and (011) components of hybrid planes, in order to distinguish from the impact on three of them, quantificationally, see Fig. 4i, j, k and l. The (111) component had the fewest epitaxial c-Si growth and the most twin-Si formation. The (011) component had the lower epitaxial c-Si growth and twin-Si formation than that of (100) component on hybrid c-Si plane. Therefore, compared with the traditional standard (111) and the above observed hybrid (111)/(100) c-Si plane in experiment, the hybrid (111)/(011) c-Si plane could have more possibility to improve the c-Si/a-Si interfacial morphology; while the latter could be realized on the side surface of textured pyramid with apex angle $\theta$ slightly smaller than 70.53°, see Fig. 4c.

## Discussion

Based on the model from c-Si/a-Si:H interfacial morphology in high-efficiency SHJ industrial devices via ex- and in situ high resolution TEM images, we performed microsecond-long low-temperature (500 K) molecular dynamic simulations, and proposed a strategy to prevent from both epitaxial c-Si growth and nanotwin formation, that the texture could have a decreased pyramid apex-angle (slightly smaller than 70.53°). The new apex-angle consists of hybrid $(111)_{0.9}/(011)_{0.1}$ plane, rather than pure (111) plane in conventional textured pyramid. The nature of the hybrid c-Si plane is low-energy c-Si/a-Si interface, which is unfavorable for either epitaxial c-Si growth or nanotwins formation. More importantly, without additional industrial preparation process, the hybrid plane could improve c-Si/a-Si:H interfacial

morphology for a-Si passivated contacts technique, and widely applied to all crystal silicon-based (single-junction and tandem) solar cells.

## Methods

### Molecular dynamics (MD) simulations

The classical MD simulations were performed by using a large-scale atomic/molecular massively parallel simulator (LAMMPS)[23]. H atoms have crucial roles in the saturation of dangling bonds in the a-Si:H layer and at the c-Si/a-Si:H interface[15,24]. Furthermore, the a-Si:H layer including the presence of appropriate amounts of H atoms is investigated[25-30]. However, the Si-H potential function shows a over-estimating melting point than that of experiment[31]. In order to describe the effect of the H atom on the pure (111) and hybrid (111)/(011) plane, we performed all atom MD simulations for c-Si/a-Si:H interfacial morphology with 10% hydrogen content at high-temperature (1000 K) for 300 ns, considering the time-consuming MD simulations (see details in Supplementary Note 2). At c-Si/a-Si:H interface, the hybrid (111)/(011) plane prevents from both epitaxial c-Si growth and nanotwin formation, which was the consistent with c-Si/a-Si interface. Therefore, considering the melting point consistent with the experiment and the calculation time consuming, all MD simulations are used with the Tersoff-mod potential to derive Si atomic interactions[32]. In order to avoid the edge effect, the periodic boundary conditions (PBC) and a time step of 2 femtoseconds (fs) were applied in the simulations. All equilibrium structures were obtained in the isothermal-isobaric (NPT) ensemble with a pressure of 1 bar at the room temperature ($T = 300$ K). The pressure was controlled by means Parrinello-Rahman Barosta. Considering sample preparation process ($T = 500$ K) of high-efficiency (PCE = 24.85%) SHJ industrial devices[11], Nosé-Hoover thermostat and Langevin method was applied for time-dependent morphological evolutions of c-Si/a-Si interface and a-Si deposition processes, with the same temperature as 500 K, respectively. The length scale of modeling systems was around 13 nm × 5 nm × 10 nm. The c-Si/a-Si interface model is composed of c-Si and a-Si with total amount of 35,000 atoms. The c-Si, twin-Si and a-Si atoms were determined by the second nearest neighbors arrangement of each central silicon atom and using Open Visualization Tool (OVITO)[33,34]. When the second nearest neighbors are arranged on a face center cubic (FCC) lattice (or hexagonal close packed (HCP) lattice), the central silicon atom was marked as c-Si (or twin-Si) atom, otherwise it was marked as a-Si atom.

### Density functional theory (DFT) calculations

All DFT calculations were obtained by employing the Vienna Ab-initio Simulation Package (VASP)[35-37] and projector augmented wave (PAW)[38] methods. The exchange correlation can be described in the form of GGA-PBE[39]. The long-range van der Waals interactions proposed by Grimme et al. (DFT-D3)[40] and the kinetic energy cutoff of electron wave functions was 500 eV. The structure of the transition state was obtained via the nudged-elastic-band (NEB) method, which calculated the minimum energy curves along a prescribed dissociation pathway[41,42]. The initial and final states were based on the optimized structures with the total energy of the system converged to less than $10^{-5}$ eV, as well as, the force on each atom converged to less than $0.01$ eV·Å$^{-1}$.

### Reporting summary

Further information on research design is available in the Nature Portfolio Reporting Summary linked to this article.

## Data availability

Source data for Fig.s 2a, b, 3 and 4i, j, k, l in the main manuscript are provided with this paper. Source data are provided with this paper.

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

## Acknowledgements

The authors are grateful for the financial support from the National Natural Science Foundation of China and Beijing Municipal Education Com-mission. Z.Z. acknowledges the support by the National Key R&D Program of China (grants 2022YFB4200200) and the National Natural Science Foundation of China (NSFC, Grants 52073005 and 22033006). Y.Z. acknowledges the support by the National Natural Science Foundation of China (NSFC, Grants 61922005 and U1930105), Beijing Natural Science Foundation (BNSF, Grants JQ20027). K.Z. acknowledges the support by the National Natural Science Foundation of China (NSFC, Grants 12074015). We also acknowledge for the financial support from Key Laboratory of Advanced Functional Materials, Education Ministry of China, and supercomputer center of Institute of Advanced Energy Materials and Devices in BJUT.

## Author contributions

F.C. carried out MD and DFT calculations. X.Q. prepared the TEM sample and executed aberration-corrected HRTEM and HAADF-STEM. Y.H. fabricated the devices. W.L., X.C., and Z.Z. assisted with the theoretical calculations. F.C., Z.Z., Y.Z., and K.Z. analyzed the experimental results and wrote the manuscript. M.Y., X.R., and F.P. assisted in the fabrication of devices. M.Q. and H.Y. provided constructive suggestions. Y.Z., Z.Z., K.Z., and X.X. led the entire project. All authors read the manuscript and contributed to the discussion of the results.

## Competing interests

The authors declare no competing interests.
