## [Peer Review File · Nature Communications]

Prediction of sub-pyramid texturing as the next step towards high efficiency silicon heterojunction solar cellsREVIEWER COMMENTS

Reviewer #1 (Remarks to the Author):

The authors proposed a new strategy to lower the formation of both epitaxial c-Si and nanotwin c-Si at c-Si/a-Si interface, via an alternative small pyramid apex-angle consisting of hybrid (111)/(011) c-Si planes. The c-Si/a-Si interface improvement for a-Si passivation contacts technique would be very interesting for a broad audience of researcher working in silicon solar cells. The nature of the atomic steps at c-Si/a-Si interface were addressed, following their previous report (nature energy 6, 194, 2021), based on a combination of MD simulations with new-found configurations from industrial silicon solar cells via atomic resolution HRTEM images. Furthermore, this work enhanced the atomic step function with new designed c-Si/a-Si interface, which is a key success factor in the efficiency improvement. The present form has sufficient results to justify the novelty of a good journal paper, and I recommend this article for publication in nature communications after minor revision.

1. The description of the first sentence of the abstract is not appropriate, please revise it. My suggestion:crystalline silicon/amorphous silicon (c-Si/a-Si) is a key success factor to approach the theoretical efficiency of Si heterojunction (SHJ) solar cell.
2. Page 3, line 77. It is said that that an intrinsic a-Si deposition ultra-thin (less than 1 nm) buffer layer formed by radio-frequency plasma enhanced chemical vapor deposition (RF-PECVD) could lower the formation of c-Si epitaxy and nanotwins. What's the difference between this intrinsic a-Si buffer layer and the conventional intrinsic a-Si under the doped a-Si layer.
3. Are the "Twin" labeled in Fig. 1d and the "Nanotwin" in Fig. 2d the same. If there are any different, could the authors let us know.
4. What is the difference between epitaxial crystalline silicon and twinned crystalline silicon in Figure S3?
5. Please explain in detail why the authors employed Tersoff-mod to describe the Si atomic interaction?
6. Compared with the density of new generated epitaxial c-Si and twin-Si on pure (111) and hybrid c-Si plane, as shown in Fig. 3a and b, why the epitaxial c-Si and twin-Si have levelled off following a period of rapid linear rise in the simulations of amorphous Si deposition simulation, as shown in 3c and d.
7. Does hybrid (100)/(011) plane have better blocking effect on c-Si epitaxial growth and nanotwin formation?

Reviewer #2 (Remarks to the Author):

In this work, the authors report on the effects of c-Si plane directions on the epitaxial-Si and nanotwin-Si formations at a-Si/c-Si interface, which is a key for achieving excellent surface passivation in SHJ solar cells, by means of MD simulation. Here the authors show that the use of hybrid plane like (111)/(100) or (111)/(110) is beneficial to suppress twin-Si, which are likely detrimental to surface passivation by MD simulation in quantitative way. These results suggest a possible way for further improvement of c-Si cells, although these results must be confirmed experimentally in the future. In my opinion, this work is worth to be published. However, the following issues should be addressed by the authors before acceptance.

(1) Effect of H

In this work, the effect of H is completely ignored, though real SHJ devices are fabricated with hydrogenated a-Si (a-Si:H) grown by PECVD or HWCVD without exception to my knowledge. In fact, a-Si:H includes few – few tens of % of H atoms, according to the literature (for example, the following references). It is understandable that the authors focus only on Si in MD simulation for simplicity. If so, the authors should state so clearly in the introduction part. In addition, if possible, please comment on what happens if H is included in MD simulation.

H. Sai et al., JAP 124 (2018) 103102

X. Ru et al., SOLMAT 215 (2020) 110643 (see, FT-IR spectra)

(2) Notation – a-Si:H or a-Si

Related to the above comment, for many years, a-Si:H (not a-Si) has been used as the abbreviation of hydrogenated amorphous silicon, in which hydrogen plays crucial roles for surface passivation. Please distinguish between a-Si:H/c-Si (real interface) and a-Si/c-Si (simulated interface).

(3) Control of apex angle

Please explain how to control the apex angle of pyramids to $> 70.53^\circ$?

(4) Fig. 3c, d

The sequence of the epi-Si/twin-Si formation is not clear. Epi-Si is transformed from a-Si after deposition? Or Epi-Si grows almost directly when Si atoms attach to the surface? This might become clearer if you can show the deposition (growth) rate of a-Si in this simulation.

(5) Correlation between the density of nanotwins and the actual defect density

It is said that the interface defect density at a-Si:H/c-Si of high-efficiency SHJ devices is in the order of 10^{10} cm^{-2} , while that of nanotwins is much higher by several orders of magnitude. Could you comment on this point?

S. Olibet et al., PRB 76 (2007) 035326

D. Adachi et al., APL 107 (2015) 081601

Reviewer #3 (Remarks to the Author):

In their manuscript "Prediction of sub-pyramid texturing as the next step towards high efficiency silicon heterojunction solar cells", Chu and co-authors investigate the impact of the crystallographic orientation at the c-Si texture surface on the growth of the amorphous silicon passivation layer in silicon heterojunction solar cells using a combination of molecular dynamics and high-resolution transmission electron microscopy. They find that hybrid surfaces with a mix of (111) and (011) c-Si planes – which according to the authors can be produced experimentally by adjusting the KOH etching procedure – show comparably low epitaxial Si growth as on the standard (111) surface, but a significant decrease of nano-twin domain formation. In the MD simulation, this finding is established both under long-duration thermal annealing at 500K (relevant for SHJ fabrication conditions) and in a simulated a-Si deposition process. Together with the optically favorable smaller apex angle, the lower concentration of potentially detrimental nano-twin domains holds the potential of better interface passivation quality and – in consequence – improved device performance.

The manuscript is concise and well written, and the conclusions regarding the dependence of epitaxial growth and twin-formation on the surface orientation are well supported by the combination of the TEM images and the MD simulations, that show a largely consistent picture regarding the atomic arrangement at the canonical and the hybrid surfaces, respectively.

On the other hand, I have two major concerns regarding the significance of the results. First of all, the specific nature of the bonds in the vicinity of the amorphous-crystalline interface is not investigated in detail, neither is the role of hydrogen discussed. In fact, hydrogen does not seem to be included in the simulation at all, even though it is supposed to play a pivotal role in the saturation of dangling bonds in the a-Si:H phase and at the a-Si:H/c-Si interface. The authors should comment on the justification for using a simple classical MD approach without hydrogen to produce the interface configuration, given that much more accurate ab initio (Born-Oppenheimer MD) schemes including the presence of appropriate amounts of hydrogen have been demonstrated in the literature [1,2], based on extensive investigation into the complex case of a-Si:H [3-6]. The change in the morphology of the c-Si surface in the presence of additional lattice planes (steps) should affect also the amount of strained or broken bonds in the a-Si:H phase and at the interface, with significant impact on the density of recombination centers in the interface region.

This leads directly to the second point of concern, namely the lack of opto-electronic device characteristics that support the claim of performance improvement related to the modified interface morphology and dominance of the recombination by nano-twin domain formation. Since the authors claim that the modified etching procedure is compatible with standard SHJ cell production, a statistically significant number of samples with modified hybrid surfaces should be produced and characterized, in order to demonstrate the advantage of the modified surface treatment, i.e., that it indeed translates into longer carrier lifetimes, which would be a significant advancement in the field. Also, in order to directly relate the nano-twin reduction to the improvement of device performance, the impact of the former on carrier lifetime needs to be expressed in a quantitative fashion.

Without the connection to the device level performance, I recommend to submit the paper to a more specialized journal focusing on microscopic morphology analysis and structure-property relations.

As a minor remark, the labels and annotations in many of the subfigures are barely readable due to the very small size.

References:

- [1] Jarolimek, K., Hazrati, E., de Groot, R.A., de Wijs, G.A.: Band offsets at the interface between crystalline and amorphous silicon from first principles. *Phys. Rev. Appl.* 8, 014026 (2017). <https://doi.org/10.1103/PhysRevApplied.8.014026>.
- [2] Czaja, P., Giusepponi, S., Gusso, M. et al. Computational characterization of a-Si:H/c-Si interfaces. *J Comput Electron* 17, 1457–1469 (2018). <https://doi.org/10.1007/s10825-018-1238-1>
- [3] Jarolimek, K., de Groot, R.A., de Wijs, G.A., Zeman, M.: First principles study of hydrogenated amorphous silicon. *Phys. Rev. B* 79, 155206 (2009). <https://doi.org/10.1103/PhysRevB.79.155206>

[4] Khomyakov, P.A., Andreoni, W., Afify, N.D., Curioni, A.: Largescale simulations of a-Si:H: the origin of midgap states revisited. *Phys. Rev. Lett.* 107, 255502 (2011). <https://doi.org/10.1103/PhysRevLett.107.255502>

[5] Legesse, M., Nolan, M., Fagas, G.: Revisiting the dependence of the optical and mobility gaps of hydrogenated amorphous silicon on hydrogen concentration. *J. Phys. Chem. C* 117(45), 23956 (2013). <https://doi.org/10.1021/jp408414f>

[6] Philippe Czaja, Massimo Celino, Simone Giusepponi, Michele Gusso, Urs Aeberhard, Ab initio study on localization and finite size effects in the structural, electronic, and optical properties of hydrogenated amorphous silicon, *Computational Materials Science* 155, 159-168 (2018), <https://doi.org/10.1016/j.commatsci.2018.08.027>.

Reviewer #1:

The authors proposed a new strategy to lower the formation of both epitaxial c-Si and nanotwin c-Si at c-Si/a-Si interface, via an alternative small pyramid apex-angle consisting of hybrid (111)/(011) c-Si planes. The c-Si/a-Si interface improvement for a-Si passivation contacts technique would be very interesting for a broad audience of researcher working in silicon solar cells. The nature of the atomic steps at c-Si/a-Si interface were addressed, following their previous report (Nature Energy 6, 194, 2021), based on a combination of MD simulations with new-found configurations from industrial silicon solar cells via atomic resolution HRTEM images. Furthermore, this work enhanced the atomic step function with new designed c-Si/a-Si interface, which is a key success factor in the efficiency improvement. The present form has sufficient results to justify the novelty of a good journal paper, and I recommend this article for publication in Nature Communications after minor revision.

We appreciate your valuable time to evaluate our work, and your comment is great helpful for improving the manuscript.

1. The description of the first sentence of the abstract is not appropriate, please revise it. My suggestion:crystalline silicon/amorphous silicon (c-Si/a-Si) is a key success factor to approach the theoretical efficiency of Si heterojunction (SHJ) solar cell.

Thanks a lot for your suggestion. We made the revision following your suggestion in the abstract section, “Over the past few decades, silicon-based solar cells have dominated the photovoltaic industry, and the interfacial morphology of crystalline silicon/hydrogenated amorphous silicon (c-Si/a-Si:H) is a key success factor to approach the theoretical efficiency of Si-based solar cells, especially Si heterojunction (SHJ) technology.”

2. Page 3, line 77. It is said that that an intrinsic a-Si deposition ultra-thin (less than 1 nm) buffer layer formed by radio-frequency plasma enhanced chemical vapor deposition (RF-PECVD) could lower the formation of c-Si epitaxy and nanotwins. What's the difference between this intrinsic a-Si buffer layer and the conventional

intrinsic a-Si under the doped a-Si layer.

Thank you for your comment. The conventional intrinsic a-Si:H layer was prepared by very-high-frequency plasma enhanced chemical vapor deposition (VHF-PECVD) with a relatively high deposition rate ($>8 \text{ \AA/s}$). However, VHF-PECVD prepared a-Si:H films often exhibit high bulk defect density and non-uniformity¹, which may lead to efficiency losses. Therefore, we introduced intrinsic a-Si:H buffer layer with a low deposition rate (approximately 2 \AA/s) grown prior to the conventional intrinsic a-Si:H layer, and the buffer layer was prepared by radio frequency (RF-) PECVD to improve c-Si surface passivation and the efficiency of SHJ solar cell. Compared with conventional intrinsic a-Si, the intrinsic a-Si:H buffer layer has high hydrogen content and large microstructure factor (R^*), defined as $R^* = I_{\text{Si-H}_2}/(I_{\text{Si-H}_2}+I_{\text{Si-H}})$, where $I_{\text{Si-H}_2}$ and $I_{\text{Si-H}}$ are peak intensities in Fourier transform infrared spectroscopy.² Therefore, the intrinsic a-Si:H buffer layer can improve the c-Si surface passivation effectively.

REFERENCES

1. J. Yang, B. Yan, G. Yue, S. Guha, Light trapping in hydrogenated amorphous and nano-crystalline silicon thin film solar cells, *MRS Proceedings* **1153**, 1153-A1113-1102 (2009).
2. Ru X, Qu M, Wang J, Ruan T, Yang M, Peng F, et al. 25.11% efficiency silicon heterojunction solar cell with low deposition rate intrinsic amorphous silicon buffer layers. *Solar Energy Materials and Solar Cells* **215**, 110643 (2020).

3. Are the “Twin” labeled in Fig. 1d and the “Nanotwin” in Fig. 2d the same. If there are any different, could the authors let us know.

Thank you for your comment. We are sorry for not labeling nanotwin at c-Si/a-Si interface clearly. The “Twin” labeled in Fig. 1d is Nanotwin, thus, we revised labeling in Fig. 1d, as shown in Fig. 1d.

Figure 1 | c-Si/a-Si:H interface structure characteristics of SHJ solar cell. **a**, Cross-section schematic image of the SHJ solar cell, from top to bottom are Ag (silver electrode), ITO (indium tin oxide), n- μ c-SiO_x (n-type microcrystal SiO_x), i-a-Si (intrinsic a-Si), n-c-Si (n-type crystal Si), i-a-Si (intrinsic a-Si), p-a-Si (p-type a-Si), ITO and Ag. **b**, SEM image showed the morphology of pyramid structure on the surface of SHJ solar cells. **c**, Low-magnification TEM image showed the cross-section of pyramid structure marked by red rectangle in Fig.1b. The green rectangle marked the flat (111) c-Si plane and the blue rectangle marked hybrid c-Si plane (111)/(100) with the atomic steps, respectively. **d,e**, HRTEM images viewed from the [01 $\bar{1}$] orientation presented the c-Si/a-Si interface with the flat (111) plane (**d**) and the atomic steps (**e**), while the related area marked by green rectangle and blue rectangle in Fig. 2c, respectively. The yellow dash line represented the (111) plane, the green dash line represented the (100) plane; the white dashed line represented c-Si surface and the orange rectangle represented the nanotwins.

4. What is the difference between epitaxial crystalline silicon and twinned crystalline silicon in Figure S3?

Thank you for your comment. There are two epitaxial mechanisms at c-Si/a-Si interface, including normal epitaxial growth (epitaxial crystalline silicon) and nanotwin growth, as shown in Fig R1a and b, respectively. All atoms maintain their normal arrangement except for the orientation. However, the embedded nanotwins are formed, when those two kinds of epitaxial layers with different orientation meet each other, as shown in Fig. R1c. The embedded nanotwin as the grain boundaries, would induce deep defect states

and leads to a short minority carrier lifetime, which was reported in our previous work.¹

Figure R1 | Geometrical atomic structure models at c-Si/a-Si interface, (a) c-Si epitaxy, (b) nanotwin and (c) embedded nanotwin, where blue balls represent crystal atoms and orange balls are twin atoms.

REFERENCES

1. Qu X, He Y, Qu M, Ruan T, Chu F, Zheng Z, *et al.* Identification of Embedded Nanotwins at c-Si/a-Si:H Interface Limiting the Performance of High-Efficiency Silicon Heterojunction Solar Cells. *Nat. Energy* **6**, 194-202 (2021).

5. Please explain in detail why the authors employed Tersoff-mod to describe the Si atomic interaction?

Thank you for your comment. The traditional Tersoff potential is one of the most widely used interatomic potentials for silicon. However, it presented poor description of melting point ($T_{\text{Tersoff}} > 2400$ K) for crystalline silicon, compared with experimental results ($T_{\text{exp.}} = 1687$ K).¹ Considering an accurate description of the temperature effect on the c-Si/a-Si interface morphology under the low-temperature preparation process (500K), we employed the improved force field with Tersoff-mod potential ($T_{\text{Tersoff-mod}} = 1681$ K), which have shown good agreement with the experimental melting point.²

REFERENCES

1. Marqués LA, Pelaz L, Hernández J, Barbolla J, Gilmer GH. Stability of defects in crystalline silicon and their role in amorphization. *Phys. Rev. B* **64**, 045214 (2001)
2. Kumagai T, Izumi S, Hara S, Sakai S. Development of Bond-Order Potentials that can Reproduce the Elastic Constants and Melting Point of Silicon for Classical Molecular Dynamics Simulation. *Comp. Mater. Sci.* **39**, 457-464 (2007).

6. Compared with the density of new generated epitaxial c-Si and twin-Si on pure (111) and hybrid c-Si plane, as shown in Fig. 3a and b, why the epitaxial c-Si and twin-Si have levelled off following a period of rapid linear rise in the simulations of amorphous Si deposition simulation, as shown in 3c and d.

Thank you for your comment. When Si atoms were deposited on the (111) plane, partial deposited silicon atoms are located on lattice sites, which lead to the nucleation of epitaxial c-Si and twin-Si, resulting in epitaxial c-Si and twin-Si increasing linearly, as shown in Fig. 3c and d. Therefore, both epitaxial c-Si and twin-Si have levelled off following a period of rapid linear rise, and it indicated nucleation-completion. However, the nuclear growth process cannot be showed due to the short time of deposition simulations. In order to investigate impact of low-temperature (500 K) on nucleation growth, we then performed microsecond-long ($\sim\mu\text{s}$) low-temperature molecular dynamic (MD) simulations, as shown in Fig 3a and b. It revealed epitaxial c-Si and twin-Si nucleation process in Fig. 3c and d, and nucleation growth in Fig 3a and b

7. Does hybrid (100)/(011) plane have better blocking effect on c-Si epitaxial growth and nanotwin formation?

Thank you for your comment. Considering nanotwin formed on the (111) plane, nanotwin growth can be blocked on the other planes (for instance, (100), (011) and hybrid (100)/(011) planes).¹ However, these planes may promote c-Si epitaxial growth, because of orientation dependent c-Si epitaxy rates ($v_{c(100)} > v_{c(011)} > v_{c(111)}$).² The epitaxial c-Si leads to reduce the passivation effect.³ Furthermore, since pyramidal textured were obtained by using KOH solution etching (100) silicon wafer in industrial

preparation process, the traditional (111) planes have the lowest etching rate compared with those of (100) and (011) planes, and thus the exposed surface is mainly composed of (111) facet. Therefore, we investigated the hybrid (111)/(100) and (111)/(011) planes rather than (100)/(011) plane as the surface of c-Si at c-Si/a-Si interface.

REFERENCES

1. Cahn RW. Twinned crystals. *Adv. Phys.* **3**, 363-445 (1954).
2. Ueno T, Showya T, Ohdomari I. Atomic scale structure of microtwins in single crystal Si grown by lateral solid phase epitaxy. *J. Appl. Phys.* **69**, 808-811 (1991).
3. Descoeurdes A, Barraud L, De Wolf S, Strahm B, Lachenal D, Guérin C, et al. Improved amorphous/crystalline silicon interface passivation by hydrogen plasma treatment. *Appl. Phys. Lett.* **99**, 123506 (2011)

Reviewer #2:

In this work, the authors report on the effects of c-Si plane directions on the epitaxial-Si and nanotwin-Si formations at a-Si/c-Si interface, which is a key for achieving excellent surface passivation in SHJ solar cells, by means of MD simulation. Here the authors show that the use of hybrid plane like (111)/(100) or (111)/(110) is beneficial to suppress twin-Si, which are likely detrimental to surface passivation by MD simulation in quantitative way. These results suggest a possible way for further improvement of c-Si cells, although these results must be confirmed experimentally in the future. In my opinion, this work is worth to be published. However, the following issues should be addressed by the authors before acceptance.

We appreciate your positive comment for this work. We believe that the new design of hybrid plane could improve c-Si/a-Si:H interfacial morphology for a-Si passivated contacts technique, and widely applied to all crystal silicon-based (single-junction and tandem) solar cells.

1. Effect of H

In this work, the effect of H is completely ignored, though real SHJ devices are fabricated with hydrogenated a-Si (a-Si:H) grown by PECVD or HWCVD without exception to my knowledge. In fact, a-Si:H includes few – few tens of % of H atoms,

according to the literature (for example, the following references). It is understandable that the authors focus only on Si in MD simulation for simplicity. If so, the authors should state so clearly in the introduction part. In addition, if possible, please comment on what happens if H is included in MD simulation.

H. Sai et al., JAP 124 (2018) 103102

X. Ru et al., SOLMAT 215 (2020) 110643 (see, FT-IR spectra)

Thank you for your valuable comment. The H atoms in a-Si:H are mainly used to passivate the c-Si/a-Si:H interface and reduce the interface defects. In consideration of the effect of SHJ preparation with low temperatures (500 K) on the interface, temperature control is a more important factor. The Si-H potential function describe the interaction of Si and H atoms, however, it has a overestimating melting point than that of experiment.¹ We therefore took the Tersoff-mod potential which is consistent with the experimental melting point ($T_{\text{Tersoff-mod}} = 1681 \text{ K}$).²

In order to describe the effect of the H atom on the pure (111) and hybrid (111)/(011) plane, we performed all atom molecular dynamics (AAMD) simulations for c-Si/a-Si:H interfacial morphology with 10% hydrogen content at high-temperature (1000 K) for 300 ns, rather than low temperatures (500 K) 1 μs in the previous manuscript, considering the time-consuming MD simulations. The growth rate of epitaxial c-Si and twin-Si at the c-Si/a-Si:H interface were obtained, as shown in Fig. R2. The c-Si epitaxy rate ($1.0 \times 10^{14} \text{ cm}^{-2} \cdot \mu\text{s}^{-1}$) with 1000 K at c-Si/a-Si:H interface was similar to that ($1.4 \times 10^{14} \text{ cm}^{-2} \cdot \mu\text{s}^{-1}$) of c-Si/a-Si interface with 500 K. The H atoms contribute to reduce the epitaxial c-Si rate, but there is not significant difference.

On the other hand, at c-Si/a-Si:H interface, comparing with the c-Si epitaxial rate ($1.0 \times 10^{14} \text{ cm}^{-2} \cdot \mu\text{s}^{-1}$) on pure (111) plane, the hybrid (111)/(011) plane presented lower epitaxial c-Si rate ($0.8 \times 10^{14} \text{ cm}^{-2} \cdot \mu\text{s}^{-1}$), which was the consistent with the conclusion in the previous manuscript. The additional H atoms with 10% content did not presented significant influence on the epitaxial c-Si on pure (111) or hybrid (111)/(011) plane. In addition, the twin-Si start reducing with the rate of $-2.1 \times 10^{14} \text{ cm}^{-2} \cdot \mu\text{s}^{-1}$ after 300 ns, and this phenomenon is consistent with that in the manuscript, which showed that the hybrid (111)/(011) plane inhibited the growth of twin-Si.

Combining the first response comment from Reviewer #3, we revised relevant discussions in the method section as following, “H atoms have crucial roles in the saturation of dangling bonds in the a-Si:H layer and at the c-Si/a-Si:H interface.^{15, 18 18} Furthermore, the a-Si:H layer including the presence of appropriate amounts of H atoms is extensively investigated.¹⁹⁻²⁴ Considering the melting point consistent with the experiment and the calculation time consuming, all MD simulations are used with the Tersoff-mod potential to derive Si atomic interactions.²⁵”

Figure R2 | Molecular dynamic simulations at c-Si/a-Si:H interface on pure (111) c-Si plane and hybrid (111)_{0.9}/(011)_{0.1} c-Si plane. The generation of both (a) epitaxial c-Si and (b) twin-Si atoms as a function of time.

REFERENCES

1. Rowland CE, Hannah DC, Demortiere A, Yang J, Cook RE, Prakapenka VB, et al. Silicon nanocrystals at elevated temperatures: retention of photoluminescence and diamond silicon to beta-silicon carbide phase transition. *ACS Nano* 8, 9219-9223 (2014).
2. Kumagai T, Izumi S, Hara S, Sakai S. Development of Bond-Order Potentials that can Reproduce the Elastic Constants and Melting Point of Silicon for Classical Molecular Dynamics Simulation. *Comp. Mater. Sci.* **39**, 457-464 (2007).
15. Ru X, Qu M, Wang J, Ruan T, Yang M, Peng F, et al. 25.11% efficiency silicon heterojunction solar cell with low deposition rate intrinsic amorphous silicon buffer layers. *Solar Energy Materials and Solar Cells* 215, 110643 (2020).
18. Sai H, Chen P.W, Hsu H.J, Matsui T, Nunomura S, Matsubara K, Impact of intrinsic amorphous silicon bilayers in silicon heterojunction solar cells. *J. Appl. Phys.* **124**,

- 103102 (2018).
19. Jarolimek K, Hazrati E, de Groot RA, de Wijs GA. Band Offsets at the Interface between Crystalline and Amorphous Silicon from First Principles. *Phys. Rev. Appl.* 8, 014026 (2017)
 20. Czaja P, Giusepponi S, Gusso M, Celino M, Aeberhard U. Computational characterization of a-Si:H/c-Si interfaces. *J. Comput. Electron.* 17, 1457-1469 (2018)
 21. Jarolimek K, de Groot RA, de Wijs GA, Zeman M. First-principles study of hydrogenated amorphous silicon. *Phys. Rev. B* 79, 155206 (2009)
 22. Khomyakov PA, Andreoni W, Afify ND, Curioni A. Large-scale simulations of a-Si:H: the origin of midgap states revisited. *Phys. Rev. Lett.* 107, 255502 (2011)
 23. Legesse M, Nolan M, Fagas G. Revisiting the Dependence of the Optical and Mobility Gaps of Hydrogenated Amorphous Silicon on Hydrogen Concentration. *J. Phys. Chem. C* 117, 23956-23963 (2013)
 24. Czaja P, Celino M, Giusepponi S, Gusso M, Aeberhard U. Ab initio study on localization and finite size effects in the structural, electronic, and optical properties of hydrogenated amorphous silicon. *Comp. Mater. Sci.* 155, 159-168 (2018)
 25. Kumagai T, Izumi S, Hara S, Sakai S. Development of Bond-Order Potentials that can Reproduce the Elastic Constants and Melting Point of Silicon for Classical Molecular Dynamics Simulation. *Comp. Mater. Sci.* 39, 457-464 (2007).

2. Notation – a-Si:H or a-Si

Related to the above comment, for many years, a-Si:H (not a-Si) has been used as the abbreviation of hydrogenated amorphous silicon, in which hydrogen plays crucial roles for surface passivation. Please distinguish between a-Si:H/c-Si (real interface) and a-Si/c-Si (simulated interface).

Thank you for your suggestion. The manuscript was modified according to the suggestions and marked with highlight.

3. Control of apex angle

Please explain how to control the apex angle of pyramids to > 70.53 degree?

Thank you for your comment. We used chemical polishing (CP) etching to achieve control apex-angle of pyramid.¹ With strong oxidizing agent (nitric acid or ozone) to oxidize the silicon pyramid surface, and silicon dioxide layer was formed, then the pyramid apex-angle of $> 70.53^\circ$ can be obtained by removing the oxide layer with hydrofluoric acid. Furthermore, based on the control of nitric acid (or ozone) concentration, the growth rate and thickness of oxide layer can be adjusted, and then the etching rate of oxide layer was tuned by controlling HF concentration, and finally the controlling of pyramid apex-angle was achieved.

REFERENCES

1. Du J, Meng F, Fu H, Sun L, Zhang L, Han A, et al. Selective rounding for pyramid peaks and valleys improves the performance of SHJ solar cells. *Energy Science & Engineering* 9, 1306-1312 (2021)

4. The sequence of the epi-Si/twin-Si formation is not clear. Epi-Si is transformed from a-Si after deposition? Or Epi-Si grows almost directly when Si atoms attach to the surface? This might become clearer If you can show the deposition (growth) rate of a-Si in this simulation.

Thank you for your comment. Following reviewer's suggestion, we made the Fig. R3, which showed densities of a-Si, epi-Si and twin-Si on pure (111), hybrid (111)_{0.9}/(100)_{0.1} and hybrid (111)_{0.9}/(011)_{0.1} plane as a function of time. At the beginning, as Si atoms were deposited on the c-Si surface, they transformed into a-Si, epi-Si and twin-Si at the same time, and both epi-Si and twin-Si started generating immediately, see the inset in Fig. R3. However, both epi-Si and twin-Si had levelled off following a period (in the first 10 ns) of rapid linear rise, which corresponded to the nucleation of epi-Si and twin-Si. After epi-Si and twin-Si reaching saturation (after 20 ns), all the deposited Si atom transformed into only a-Si, and the a-Si presents linear growth, where the a-Si growth rate is the same as the Si deposition rate.

Figure R3 | The densities of a-Si, epitaxial c-Si and twin-Si on (a) pure (111), (b) hybrid (111)_{0.9}/(100)_{0.1} and (c) hybrid (111)_{0.9}/(011)_{0.1} plane as a function of time.

5. Correlation between the density of nanotwins and the actual defect density

It is said that the interface defect density at a-Si:H/c-Si of high-efficiency SHJ devices is in the order of 10^{10} cm^{-2} , while that of nanotwins is much higher by several orders of magnitude. Could you comment on this point?

S. Olibet et al., PRB 76 (2007) 035326

D. Adachi et al., APL 107 (2015) 081601

Thank you for your comment. We are sorry for not clearly expressing the relevant discussion about density of nanotwins. In a-Si:H layer, dangling bond (DB) states in surface and interface of c-Si and nanotwin is the dominant mechanism for nonradiative recombination at room temperature. The DB states lead to defect density of about 10^{10} cm^{-2} at c-Si/a-Si:H interface of high-efficiency SHJ devices. In our work, we can not obtain defect density at c-Si/a-Si:H interface duo to molecular dynamics simulations. Therefore, we observed and showed epitaxial c-Si and nanotwin atomic density to describe defect density at c-Si/a-Si interface. The nanotwin and epitaxial c-Si consist of thousands of Si atoms, which may introduce a few BD states, and the nanotwin atomic density is several orders of magnitude more than interface defect density at interface. We added relevant discussions in the manuscript as following, “Comparing with interface defect density at c-Si/a-Si:H of high-efficiency SHJ devices (about 10^{10} cm^{-2})^{36, 37}, the interfacial epitaxial c-Si and nanotwin atomic density is several orders of magnitude more than interface defect density, in consideration of the nanotwin or epitaxial c-Si consisting of thousands of Si atoms only introduce a few defect states.”

REFERENCES

36. Geissbühler J, Werner J, Martin de Nicolas S, Barraud L, Hessler-Wyser A, Despeisse M, et al. 22.5% efficient silicon heterojunction solar cell with molybdenum oxide hole collector. *Appl. Phys. Lett.* **107**, 081601 (2015)
37. Olibet S, Vallat-Sauvain E, Ballif C. Model for a-Si:H/c-Si interface recombination based on the amphoteric nature of silicon dangling bonds. *Phys. Rev. B* **76**, 035326 (2007)

Reviewer #3:

In their manuscript “Prediction of sub-pyramid texturing as the next step towards high efficiency silicon heterojunction solar cells”, Chu and co-authors investigate the impact of the crystallographic orientation at the c-Si texture surface on the growth of the amorphous silicon passivation layer in silicon heterojunction solar cells using a combination of molecular dynamics and high-resolution transmission electron microscopy. They find that hybrid surfaces with a mix of (111) and (011) c-Si planes – which according to the authors can be produced experimentally by adjusting the KOH etching procedure – show comparably low epitaxial Si growth as on the standard (111) surface, but a significant decrease of nano-twin domain formation. In the MD simulation, this finding is established both under long-duration thermal annealing at 500K (relevant for SHJ fabrication conditions) and in a simulated a-Si deposition process. Together with the optically favorable smaller apex angle, the lower concentration of potentially detrimental nano-twin domains holds the potential of better interface passivation quality and – in consequence – improved device performance.

The manuscript is concise and well written, and the conclusions regarding the dependence of epitaxial growth and twin-formation on the surface orientation are well supported by the combination of the TEM images and the MD simulations, that show a largely consistent picture regarding the atomic arrangement at the canonical and the hybrid surfaces, respectively.

Thank you for your positive comments, which greatly encourage us to continue working on the interfacial study in silicon solar cells, and we will improve our work according to your comments.

1. On the other hand, I have two major concerns regarding the significance of the

results. First of all, the specific nature of the bonds in the vicinity of the amorphous-crystalline interface is not investigated in detail, neither is the role of hydrogen discussed. In fact, hydrogen does not seem to be included in the simulation at all, even though it is supposed to play a pivotal role in the saturation of dangling bonds in the *a*-Si:H phase and at the *a*-Si:H/*c*-Si interface. The authors should comment on the justification for using a simple classical MD approach without hydrogen to produce the interface configuration, given that much more accurate *ab initio* (Born-Oppenheimer MD) schemes including the presence of appropriate amounts of hydrogen have been demonstrated in the literature [1,2], based on extensive investigation into the complex case of *a*-Si:H [3-6]. The change in the morphology of the *c*-Si surface in the presence of additional lattice planes (steps) should affect also the amount of strained or broken bonds in the *a*-Si:H phase and at the interface, with significant impact on the density of recombination centers in the interface region.

References:

- [1] Jarolimek, K., Hazrati, E., de Groot, R.A., de Wijs, G.A.: Band offsets at the interface between crystalline and amorphous silicon from first principles. *Phys. Rev. Appl.* 8, 014026 (2017).
- [2] Czaja, P., Giusepponi, S., Gusso, M. et al. Computational characterization of *a*-Si:H/*c*-Si interfaces. *J Comput Electron* 17, 1457–1469 (2018).
- [3] Jarolimek, K., de Groot, R.A., de Wijs, G.A., Zeman, M.: First principles study of hydrogenated amorphous silicon. *Phys. Rev. B* 79, 155206 (2009).
- [4] Khomyakov, P.A., Andreoni, W., Afify, N.D., Curioni, A.: Largescale simulations of *a*-Si:H: the origin of midgap states revisited. *Phys. Rev. Lett.* 107, 255502 (2011).
- [5] Legesse, M., Nolan, M., Fagas, G.: Revisiting the dependence of the optical and mobility gaps of hydrogenated amorphous silicon on hydrogen concentration. *J. Phys. Chem. C* 117(45), 23956 (2013).
- [6] Philippe Czaja, Massimo Celino, Simone Giusepponi, Michele Gusso, Urs Aeberhard, *Ab initio study on localization and finite size effects in the structural, electronic, and optical properties of hydrogenated amorphous silicon*, *Computational Materials Science* 155, 159-168 (2018).

Thank you for your comment. We are sorry for that we have not investigated the nature of the bonds in the vicinity of the *a*-Si interface. We obtained the interface morphology

of SHJ solar cells by employing all atom molecular dynamics (AAMD) based on Newton's equations, rather than ab initio molecular dynamics (AIMD), and the former cannot provide the information about the interfacial electronic states and dangling bond states. The H atoms have a crucial role in the saturation of dangling bonds in the a-Si:H layer and at the c-Si/a-Si:H interface, and the importance of H atoms in a-Si:H could be investigated by AIMD. However, the AIMD can only calculate small modeling systems (~ 1 nm) containing hundreds of atoms within a short time at picosecond (\sim ps) level. Considering the nanotwin and epitaxial c-Si models containing hundreds of atoms, the AIMD is unable to describe the mesoscopic morphology with long time at microsecond level for c-Si/a-Si:H interface in this work. Therefore, we employed AAMD in this work, which simulated modeling systems with tens of thousands of atoms, in order to obtain the interface morphology (~ 50 nm) with time dependent evolution (~ 1 μ s).

We discussed the reasons for applying Si potential rather than Si-H potential in detail in the first response comment from reviewer #2. The main three reasons are as following, (i) The melting points of Si-H potential is overestimated comparing with experimental results; therefore, if taking Si-H potential in the MD simulations, the c-Si/a-Si interface simulations may not be with accuracy. (ii) The additional H atoms significantly increases the computational time, considering the force field with extended Tersoff-mod potential with three-body interactions. (iii) The additional H atoms have no significant influence on the epitaxial trend on pure (111) and hybrid (111)/(011) plane, see Fig. R4 as below.

In order to describe the effect of the H atom on the pure (111) and hybrid (111)/(011) plane, we performed molecular dynamics simulations for c-Si/a-Si:H interfacial morphology with 10% hydrogen content at high-temperature (1000 K) for 300 ns, rather than low temperatures (500 K) 1 μ s in the previous manuscript, considering the time-consuming MD simulations. The growth rate of epitaxial c-Si and twin-Si at the c-Si/a-Si:H interface were obtained. The c-Si epitaxy rate (1.0×10^{14} $\text{cm}^{-2} \cdot \mu\text{s}^{-1}$) with 1000 K at c-Si/a-Si:H interface was similar to that (1.4×10^{14} $\text{cm}^{-2} \cdot \mu\text{s}^{-1}$) of c-Si/a-Si interface with 500 K. The H atoms contribute to reduce the epitaxial c-Si rate, but there is not significant difference.

Figure R4 | Molecular dynamic simulations at c-Si/a-Si:H interface on pure (111) c-Si plane and hybrid (111)_{0.9}/(011)_{0.1} c-Si plane. The generation of both (a) epitaxial c-Si and (b) twin-Si atoms as a function of time.

Moreover, to explore the effect of hybrid plane on the strain or the number of broken bonds in the a-Si:H layer and at c-Si/a-Si:H interface, we calculated the surface dangling bond density on the pure (111) and hybrid (111)/(011) planes, respectively. The surface dangling bond density of pure (111) plane is $7.8 \times 10^{14} \text{ cm}^{-2}$, and that of hybrid plane is $8.5 \times 10^{14} \text{ cm}^{-2}$. It implied that the change of the c-Si surfaces has no significant effect on interfacial states.

Furthermore, low-temperature deposition and annealing treatment release the interface stress, and interface with the hybrid plane will not generate additional strain. We added relevant discussions in the method section as following, “H atoms have crucial roles in the saturation of dangling bonds in the a-Si:H layer and at the c-Si/a-Si:H interface.^{15, 18} Furthermore, the a-Si:H layer including the presence of appropriate amounts of H atoms is extensively investigated.¹⁹⁻²⁴ Considering the melting point consistent with the experiment and the calculation time consuming, all MD simulations are used with the Tersoff-mod potential to derive Si atomic interactions.²⁵”

REFERENCES

15. Ru X, Qu M, Wang J, Ruan T, Yang M, Peng F, et al. 25.11% efficiency silicon heterojunction solar cell with low deposition rate intrinsic amorphous silicon buffer

- layers. *Solar Energy Materials and Solar Cells* 215, 110643 (2020).
18. Sai H, Chen P.W, Hsu H.J, Matsui T, Nunomura S, Matsubara K, Impact of intrinsic amorphous silicon bilayers in silicon heterojunction solar cells. *J. Appl. Phys.* **124**, 103102 (2018).
 19. Jarolimek K, Hazrati E, de Groot RA, de Wijs GA. Band Offsets at the Interface between Crystalline and Amorphous Silicon from First Principles. *Phys. Rev. Appl.* **8**, 014026 (2017)
 20. Czaja P, Giusepponi S, Gusso M, Celino M, Aeberhard U. Computational characterization of a-Si:H/c-Si interfaces. *J. Comput. Electron.* **17**, 1457-1469 (2018)
 21. Jarolimek K, de Groot RA, de Wijs GA, Zeman M. First-principles study of hydrogenated amorphous silicon. *Phys. Rev. B* **79**, 155206 (2009)
 22. Khomyakov PA, Andreoni W, Afify ND, Curioni A. Large-scale simulations of a-Si:H: the origin of midgap states revisited. *Phys. Rev. Lett.* **107**, 255502 (2011)
 23. Legesse M, Nolan M, Fagas G. Revisiting the Dependence of the Optical and Mobility Gaps of Hydrogenated Amorphous Silicon on Hydrogen Concentration. *J. Phys. Chem. C* **117**, 23956-23963 (2013)
 24. Czaja P, Celino M, Giusepponi S, Gusso M, Aeberhard U. Ab initio study on localization and finite size effects in the structural, electronic, and optical properties of hydrogenated amorphous silicon. *Comp. Mater. Sci.* **155**, 159-168 (2018)
 25. Kumagai T, Izumi S, Hara S, Sakai S. Development of Bond-Order Potentials that can Reproduce the Elastic Constants and Melting Point of Silicon for Classical Molecular Dynamics Simulation. *Comp. Mater. Sci.* **39**, 457-464 (2007).

2. This leads directly to the second point of concern, namely the lack of opto-electronic device characteristics that support the claim of performance improvement related to the modified interface morphology and dominance of the recombination by nano-twin domain formation. Since the authors claim that the modified etching procedure is compatible with standard SHJ cell production, a statistically significant number of samples with modified hybrid surfaces should be produced and characterized, in order to demonstrate the advantage of the modified surface treatment, i.e., that it indeed translates into longer carrier lifetimes, which would be a significant advancement in

the field. Also, in order to directly relate the nano-twin reduction to the improvement of device performance, the impact of the former on carrier lifetime needs to be expressed in a quantitative fashion.

Surface texturing of monocrystalline silicon is known as one of the best methods to reduce reflection losses and to increase light trapping and light absorption probability. The decreased pyramid apex-angle could be prepared by changing the concentrations of potassium hydroxide (KOH) and isopropyl alcohol (IPA) and controlling the etching process time. For instance, the statistical apex-angle of 69.98° were demonstrated under the etching conditions from a solution prepared with 20 wt% KOH, 3 wt% IPA and etching time of 20 min, which was slightly smaller than standard apex-angle (70.53°)¹. The texturing pyramid with decreased apex-angle consisting of hybrid (111)/(011) plane could be prepared. However, currently, the process adjustment will cost too much to demonstrate the modified hybrid surfaces in industrial production line. The laboratorial demonstration might hardly base on the level of such efficient SHJ devices. The new production line for pilot scale testing is under planning in LONGi Central R&D Institute, in order to achieve the higher efficient silicon solar cells, which can demonstrate the industrial hybrid (111)/(011) plane SHJ devices in the future. In this work, we addressed the new designed apex-angle consists of hybrid (111)/(011) c-Si planes, rather than pure (111) planes in conventional texture pyramid, with the mechanism for preventing form both epitaxial c-Si growth and nanotwins formation, considering low-energy c-Si/a-Si hybrid interface. The hybrid c-Si plane could improve c-Si/a-Si:H interfacial morphology for a-Si passivated contacts technique, and wide-applied for all silicon-based (single-junction and tandem) solar cells as well.

REFERENCES

1. Al-Husseini AM, Lahlouh B. Influence of pyramid size on reflectivity of silicon surfaces textured using an alkaline etchant. *Bulletin of Materials Science* 42, 152 (2019).

3. Without the connection to the device level performance, I recommend to submit the paper to a more specialized journal focusing on microscopic morphology analysis and structure-property relations.

The silicon-based solar cells have dominated the photovoltaic industry, and the interfacial morphology of c-Si/a-Si:H is a key success factor to achieve higher efficiency of Si-based passivated contact solar cells, for instance, SHJ, heterojunction back-contact (HBC), and tunnel oxide passivated contact (TOPCon) technologies. One of the major issues is the unexpected c-Si epitaxial growth and nanotwins formation at c-Si/a-Si:H interface. By combining the microsecond-long low-temperature (500 K) MD simulations with atomic-resolution STEM and in-situ high resolution TEM, we obtained new finding (large texturing pyramid apex-angle consisting of hybrid (111)/(100) c-Si planes) to prevent from epitaxial c-Si in high-efficiency SHJ devices, then we proposed a new strategy (small apex-angle consists of hybrid (111)/(011) c-Si planes) to prevent from both epitaxial c-Si growth and nanotwin formation in SHJ solar cells. The new hybrid (111)/(011) c-Si plane could improve c-Si/a-Si:H interfacial morphology for Si-based passivated contacts technology, and wide-applied for many commercialized Si solar cells.

4. As a minor remark, the labels and annotations in many of the subfigures are barely readable due to the very small size.

Thank you for your suggestion. The labels and annotations in subfigures were enlarged in the revised version.

REVIEWERS' COMMENTS

Reviewer #1 (Remarks to the Author):

The authors have responded to the comments addressed in the last review. The overall responses are acceptable. I have no further questions and recommend publication.

Reviewer #2 (Remarks to the Author):

The authors replied to the comments almost adequately. However, their replies are not sufficiently reflected into the revised manuscript.

(1) Effect of H

In the response letter, the authors explain the effect of including H in their simulation and the limitation in detail. They confirmed that the additional H atoms with 10% content does not present significant influence on Si epitaxy. In contrast, their revision regarding this point is very limited and not satisfactory. They should do so also in the manuscript, to justify that their simplification (not including H atoms) is valid in a clearer manner. They can do so by showing them as supporting information.

(2) 26.5%

The record efficiency in SHJ cells is now 26.8%, which was reported by LONGi Solar last November. Please update it.

At 26.81%, LONGi sets a new world record efficiency for silicon solar cells - LONGi

(3) Typo

I found there are many typos. Please check the manuscript carefully and correct them.

Reviewer #3 (Remarks to the Author):

In their revision, the authors thoroughly addressed all of the reviewers concerns and applied the necessary amendments to the manuscript. I therefore support publication of the work in its present form.

Reviewer #1:

The authors have responded to the comments addressed in the last review. The overall responses are acceptable. I have no further questions and recommend publication.

We appreciate your positive comment for this work, and we expect the hybrid c-Si plane could improve c-Si/a-Si:H interfacial morphology for a-Si passivated contacts technique, and wide-applied for all silicon-based solar cells in the future.

Reviewer #2:

The authors replied to the comments almost adequately. However, their replies are not sufficiently reflected into the revised manuscript.

Thank you again for your positive comments and valuable suggestions to improve our manuscript.

(1) Effect of H

In the response letter, the authors explain the effect of including H in their simulation and the limitation in detail. They confirmed that the additional H atoms with 10% content does not present significant influence on Si epitaxy. In contrast, their revision regarding this point is very limited and not satisfactory. They should do so also in the manuscript, to justify that their simplification (not including H atoms) is valid in a clearer manner. They can do so by showing them as supporting information.

Thank you for your suggestion. We revised the main text following your suggestion in the new version as following “However, the Si-H potential function shows a overestimating melting point than that of experiment.³⁰ In order to describe the effect of the H atom on the pure (111) and hybrid (111)/(011) plane, we performed all atom MD simulations for c-Si/a-Si:H interfacial morphology with 10% hydrogen content at high-temperature (1000 K) for 300 ns, considering the time-consuming MD simulations, see Section S2 in SI. At c-Si/a-Si:H interface, the hybrid (111)/(011) plane prevents

from both epitaxial c-Si growth and nanotwin formation, which was the consistent with c-Si/a-Si interface.”

References

30 Rowland CE, et al. Silicon nanocrystals at elevated temperatures: retention of photoluminescence and diamond silicon to beta-silicon carbide phase transition. ACS Nano 8, 9219-9223 (2014).

(2) 26.5%

The record efficiency in SHJ cells is now 26.8%, which was reported by LONGi Solar last November. Please update it.

Thanks a lot for your suggestion. We updated the related discussions following your suggestion in the introduction section.

(3) Typo

I found there are many typos. Please check the manuscript carefully and correct them.

We are sorry for those typing mistakes, and we corrected them in the revised version.

Reviewer #3:

In their revision, the authors thoroughly addressed all of the reviewers concerns and applied the necessary amendments to the manuscript. I therefore support publication of the work in its present form.

We appreciate your positive comment for this work. We revealed suppression effect of epitaxial c-Si growth and nanotwin formation in hybrid c-Si interfacial morphology, and expect the hybrid c-Si plane could be a promising candidate with wide-applied for the silicon-based solar cells in the future.